# Obesity, Pregnancy and the Social Contract with Today’s Adolescents

**DOI:** 10.3390/nu14173550

**Published:** 2022-08-28

**Authors:** Cristiana Berti, Shirin Elahi, Patrick Catalano, Zulfiqar A. Bhutta, Michael B. Krawinkel, Francesca Parisi, Carlo Agostoni, Irene Cetin, Mark Hanson

**Affiliations:** 1Fondazione IRCCS Ca’ Granda Ospedale Maggiore Policlinico, Pediatric Unit, 20122 Milan, Italy; 2Director Corset Economy, London NW7 3TD, UK; 3Mother Infant Research Institute, Tufts University School of Medicine, Boston 02111, MA, USA; 4Centre for Global Child Health, The Hospital for Sick Children, Toronto, ON M5G 0A4, Canada; 5Center of Excellence in Women and Child Health, The Aga Khan University, Karachi 74800, Pakistan; 6Institute of Nutritional Sciences—International Nutrition, Justus-Liebig-University, 35392 Giessen, Germany; 7Department of Woman, Mother and Neonate, “V. Buzzi” Children Hospital, ASST Fatebenefratelli Sacco, 20154 Milan, Italy; 8Department of Clinical Sciences and Community Health, University of Milan, 20122 Milan, Italy; 9Department of Biomedical and Clinical Sciences, School of Medicine, University of Milan, 20154 Milan, Italy; 10Institute of Developmental Sciences and NIHR Biomedical Research Centre, University of Southampton and University Hospital Southampton, Southampton SO17 1BJ, UK

**Keywords:** adolescents, obesity, pregnancy, parental obesity, nutrition, social context

## Abstract

Adolescent health and well-being are of great concern worldwide, and adolescents encounter particular challenges, vulnerabilities and constraints. The dual challenges of adolescent parenthood and obesity are of public health relevance because of the life-altering health and socioeconomic effects on both the parents and the offspring. Prevention and treatment strategies at the individual and population levels have not been successful in the long term, suggesting that adolescent pregnancy and obesity cannot be managed by more of the same. Here, we view adolescent obese pregnancy through the lens of the social contract with youth. The disruption of this contract is faced by today’s adolescents, with work, social and economic dilemmas which perpetuate socioeconomic and health inequities across generations. The lack of employment, education and social opportunities, together with obesogenic settings, increase vulnerability and exposure to lifelong health risks, affecting their offspring’s life chances too. To break such vicious circles of disadvantage and achieve sustainable solutions in real-world settings, strong efforts on the part of policymakers, healthcare providers and the community must be oriented towards guaranteeing equity and healthy nutrition and environments for today’s adolescents. The involvement of adolescents themselves in developing such programs is paramount, not only so that they feel a sense of agency but also to better meet their real life needs.

## 1. Introduction

Adolescents’ health and well-being are of great concern worldwide [1]. In 2019, a coalition of governments, United Nations agencies, non-governmental organizations and academic institutions proposed a call to action for adolescent well-being, working closely with adolescents and young people themselves for a more concerted and collaborative approach to their well-being [2,3]. Several countries targeted the improvement of food environments with actions directed to adolescents, as summarized by Al-Jawaldeh and colleagues [4]. Adolescents were mentioned in 12 of the Sustainable Development Goals indicators relevant to health, including those associated with nutrition, reproductive health, sexual and intimate partner violence, child marriage, education and employment [5]. Investments in this population group are emphasized to deliver a “triple dividend”, i.e., improving their health now, enhancing it throughout their later life course and contributing to the health of future generations [6]. 

Despite these initiatives, the health and well-being of adolescents as a population group is still severely challenged. 

The rate of adolescents affected by overweight or obesity more than doubled to almost one in five from 1990 to 2016 [7]. Such increases are mostly the result of changes in the food environment and lifestyle [8]. Obesity development depends on the relationship between a person and the background, that is, individual physiology and behavior are shaped by strong social and environment factors. Obesity represents a thorny health issue for policymakers, it being a condition that relies on complex and interrelated biological, genetic, social, environmental and behavioral determinants and exerts damaging lifelong effects [8]. In genetically susceptible individuals there is a continuous development of obesity from childhood into adolescence [9,10]. When occurring in adolescence, obesity may have major implications for both the affected adolescent and the society [11]. It is associated with adolescent metabolic syndrome and a higher risk of obesity and non-communicable diseases (NCDs) in adulthood [8,9]. Furthermore, several psychological problems (i.e., body weight stigma, poor self-esteem, eating disorders, internet addiction, etc.) affect the social life and well-being of adolescents with obesity by contributing to addictive behaviors that may lead to a vicious poor self-management cycle (i.e., social isolation, avoidance of healthcare services, decreased physical activity, increased weight gain), which creates additional barriers to healthy behavior changes [8,12,13,14,15]. Overall, obesity causes multiple medical, psychological and social co-morbidities, leading to a reduced quality of life, unemployment, increased social disadvantagesand, ultimately, to the transmission of risks across generations [8,11]. Figure 1 illustrates the complex network of the main causes and consequences of obesity in adolescence. 

Obesity prevention and treatment strategies—both at the individual and population level—are not working in the long term. Lifestyle and behavioral interventions aimed at reducing calorie intake and increasing energy expenditure have limited effectiveness. Despite being the responsibility of an individual, behavioral changes can be ineffective in the context of a lack of supportive policies in sectors such as health, agriculture, transport, urban planning, environment, food processing and marketing and education. Approaches that combine individual interventions with changes in the environment and society are paramount [8]. 

Each year, 12 million girls aged 15–19 years and almost 1 million girls under 15 years give birth [16]. Pregnancy in these adolescents carries a higher risk of complications than it does in older women [17,18,19]. It is associated with preeclampsia and eclampsia, preterm delivery, a low birth weight and increased neonatal morbidity and mortality [20,21,22,23]. Furthermore, adolescence motherhood may have important life-altering implications and socioeconomic costs for the adolescent, her child and her family [24]. By impacting schooling, social connections, employment opportunities, market participation and dependence on state welfare programs, adolescent childbearing holds back the personal development of the girl and reduces her educational attainment and lifetime earnings, thereby nourishing a cycle of poverty for the mother–child dyad [18].

The Coronavirus disease (COVID-19) pandemic has worsened the existing threats faced by young people, with long-lasting effects on their future health and well-being, and has exacerbated pre-existing inequalities [25,26]. School closures were widely implemented across the globe during 2020, affecting over 220 million young people and having negative impacts on their opportunities for learning, critical thinking and the development of their social skills, with likely affected their employment prospects and wider contributions to society [27]. The loss of sexual and reproductive health services due to restrictions between March and August 2020 led to increased unplanned pregnancies and unsafe abortions for approximately an extra 1.3 million and 1.2 million, respectively [28]. Restrictive measures have impacted the food environment negatively and have increased food insecurity and physical inactivity, thus contributing to a raised risk for overweight and obesity among young people [7,29,30,31].

Today’s adolescents are living through “socio-historically situated intersecting crises”, including precarity and climate change [26,32]. By challenging human health and well-being with current and future global threats to weather, ecosystems and human systems, climate change is amplifying health and social inequities [33]. Climate anxiety and pessimistic beliefs about the future (e.g., *humanity is doomed; they will not have access to the same opportunities their parents had; security is threatened*; etc.), along with emotional dissatisfaction with government responses, are widespread among young people and impact their daily functioning and the risk of mental health problems [34]. Pathophysiological consequences of heat exposure are documented, including increased risks of interpersonal and collective violence [35]. Although difficult to be quantified, climate change-related pathways may alter global food production by affecting the quantity, quality and affordability of foods, thus exacerbating nutrient deficiencies, obesity and vulnerability among the most food-insecure populations [36].

Taken together these challenges can be viewed as the destruction of the social contract with young people. As defined by Loewe and colleagues [37], the social contract is “*the entirety of explicit or implicit agreements between all relevant societal groups and the sovereign (i.e., the government or any other actor in power), defining their rights and obligations toward each other*”. According to this contract, governments must guarantee protection, the provision of basic services and participation in political decision-making processes on different levels. People must exert their roles as elected officials, employers, parents, customers and citizens to influence the societal norms and institutional policies of worksites, schools, food retailers and communities [36]. Adolescents are expected to be not only the beneficiaries but also the central actors in driving the transformative change needed to advance healthy, safe and sustainable food systems and diets [7].

To date, national policymakers and the international community have failed to protect and listen to young people, and there are continuing gaps in effective programs to support healthy adolescent growth, nutrition and development [7], as well as education and social justice. Governments tend to place the responsibility for choices about healthy behaviors on citizens, without facilitating policies aimed at addressing the underlying structural determinants of such choices or taking into account that population groups such as adolescents are often unable to adopt them. As Loewe and colleagues [38] state:“*When individuals do not realistically have a choice, whether as a result of lack of capability or opportunity, or of demotivating socioeconomic contexts, this reflects the failure of society in providing those aspects of life which permit choice*”. Whilst governments may expect citizens—including adolescents, many of whom are too young to vote—to fulfill their side of the social contract in terms of their social obligations, the present situation makes it impossible for them to do so. 

The dual challenges of adolescent parenthood and obesity are of great public health relevance because of the life-altering health and socioeconomic effects on both the parents and the offspring. Here, we adopt a new perspective on the issues faced by adolescents by viewing the challenge of adolescent pregnancy against the increasingly common background of obesity in an attempt to provide policymakers, practitioners and academics with suggestions for preventive nutrition-based strategies to break the intergenerational cycle of malnutrition and NCDs. These issues are addressed according to the following framing. First, we summarize the evidence about the interactions between these conditions, that is, what is known about the short- and long-term consequences of both obesity and childbearing during adolescence, and what are the opportunities and challenges for progress in establishing healthy behaviors of parents-to-be to prevent both obesity and pregnancy during adolescence? Then, we argue that the causes of these simultaneous challenges lie in a broken social contract with young people. We explore the insights this perspective reveals through a series of dilemmas—skills, employment, economic and social—which are challenging to both the functions of governments and the responsibilities of adolescents as citizens and future parents at this critical time in their lives. This affects not only the adolescents but society at large, thereby exacerbating the breaks in the social contract even further.

## 2. The Challenges of Adolescent Pregnancy with Obesity

Adolescent motherhood is of policy and public health relevance globally because of the risk of health complications and the life-altering and socioeconomic implications for the adolescent–offspring dyad as well [39]. Many adolescent girls enter pregnancy malnourished and anemic, which greatly increases the risk of adverse pregnancy outcomes and the likelihood of stunting in their child [1,40,41]. Adolescents are more likely to consume micronutrient-poor, energy-dense diets that are rich in fat and added sugar [19,42], which represent a risk for weight gain and obesity [8]. During pregnancy, ultra-processed food intake may be a risk factor for adverse maternal weight outcomes such as greater gestational weight gain (GWG), postpartum weight retention and inflammation [43]. In contrast, dietary patterns with a higher intake of fruits, vegetables, legumes, whole grains and fish are associated with a decreased likelihood of adverse pregnancy and birth outcomes [44]. Obesity in women of reproductive age, including adolescents, is increasing in prevalence, with current estimates indicating that, by 2025, more than 21% of women worldwide will be obese [45]. A recent pooled-analysis of 2416 population-based studies, including 31.5 million people aged 5–19 years, evidenced that the global age-standardized prevalence of obesity increased from 0.7% in 1975 to 5.6% in 2016 among girls [46]. Data from the United States National Vital Statistics System found that the percentage of women with pre-pregnancy obesity rose 13% for women under the age of 20 from2016 to 2019 [47]. A report based on data from Demographic and Health Surveys regarding nutritional trends among adolescents from 2000 to 2017, which were conducted in 65 low-income and 22 middle-income countries, showed that girls were more than 10% overweight and obese in almost half of the countries, with overweightness and obesity being a larger problem than thinness and childbearing during adolescence being prevalent across all countries [41].

### 2.1. Why Does it Matter? The Triple Threat: Current Health, Future Adult Health and Health of the Next Generation 

Obesity and pregnancy represent a combination that may create interconnected challenges for the health of the next generation. As summarized in Table 1 [48,49,50,51,52,53], women of reproductive age, including adolescents, with obesity during pregnancy and their offspring are much more likely than other women and newborns to experience a range of pregnancy/obstetric complications and adverse neonatal outcomes. The underlying subclinical metabolic disturbances in pregnancy with obesity are under investigation. Insulin resistance, elevated levels of plasma triglycerides, cholesterol and leptin and chronic low-grade inflammation are thought to exert adverse effects [54].

Women and girls with pre-gravid overweightness and obesity do have significant risks for adverse pregnancy outcomes, including gestational diabetes (GDM) and preeclampsia [50]. Pre-gestational and gestational diabetes have been shown to have a high association with adverse maternal and fetal outcomes, including congenital abnormalities and peri-natal deaths [53]. In pregnancies complicated by diabetes, altered maternal lipid and amino acid metabolism, in addition to hyperglycemia, likely constitute a risk for macrosomia. Relating to fetal growth, in overweight and obese pregnancy, maternal pre-gravid BMI is the primary issue in contrast with normal or underweight mothers, where excessive GWG is more associated with fetal overgrowth and adiposity [51,55]. Excessive GWG represents a major determinant for maternal post-partum weight retention and increases the risk of cesarean delivery and preterm birth. It is worth keeping in mind that the Institute of Medicine recommendations focus on adults. Todate, GWG references to optimize pregnancy outcomes for pregnant adolescents do not exist, as in 2009, there werenot enough data to provide specific recommendations [56]. So, the values for adult women are considered valid for girls as well. 

Infants born to mothers with obesity are at an increased risk for both neonatal morbidity and long-term *sequelae* related to the abnormal intrauterine metabolic environment [51]. Body composition measures in neonates from normo-glycemic pregnancies of pre-gravid overweight or obese women versus lean or average-weight women showed that overweight or obese women gave birth to heavier neonates, and the increase in birthweight was attributable primarily to an increased adiposity [57]. Obesity in earlypregnancy likely increases the risk of obesity and metabolic syndrome in the offspring, independently of maternal GDM or excessive GWG [49]. As maternal pre-gravid BMI seems to modify the relationship between excessive GWG and offspring overweight, pre-gestational BMI needs to be considered for current GWG recommendations, based on pre-gravid BMI categories [58]. Nevertheless, the desirable degree of GWG remains a matter of debate. Considering that intrauterine exposure to diabetes *per se* conveys the risk for developing diabetes and obesity in youth, the combination of maternal obesity and GDM may have a greater impact on fetal growth than either alone [52]. GDM and macrosomia induce metabolic abnormalities, altered hypothalamic neuropeptide production, altered antioxidant status and disrupted immune systems, promoting potential obesity and metabolic syndrome in the offspring [59].

Shifts in the gut microbiota composition (i.e., reduced levels of *Bifidobacterium* spp., higher levels of *Staphylococcus* and *Enterobacteriaceae*, etc.) and lower microbial diversity during pregnancy have been observed in obese women compared to lean women. Alterations in maternal microbiota and dysbiosis might indirectly influence the fetal development and be transmitted to the offspring, thus favoring altered colonization patterns in the neonate, which are likely linked to an increased risk of NCDs [60]. Pregnant women with obesity may have derangements in one-carbon metabolism and gut dysbiosis owing to the high intake of nutritiously poor foods and a chronic systemic inflammatory state. Low folate and vitamin B12 likely coincide with the decreased presence of B vitamin-producing bacteria and the increased presence of inflammatory-associated bacteria, which are risk factors for the disruption of microbiota formation [61].

Beyond the medical threats, adolescents are highly vulnerable to psychosocial models relying on family and peer relationships and social media that may shape their dietary practices and lifestyles, potentially leading to alcohol drinking and smoking [62], which may result in detrimental consequences for both the maternal nutritional status and the overall pregnancy outcome [63,64]. All these conditions may be exacerbated by the psychological and emotional stresses of childbearing and parenting put on the adolescent due to novel responsibilities, adjustments in lifestyle and family dynamics and changes in social networks [65]. Moreover, adolescent pregnancy increases disadvantages for girls by disrupting education, limiting life chances (e.g., employment) and perpetuating the cycle of poverty [1].

### 2.2. Back to Basics 

Enhancing positive environmental exposures during early life offers important potential for the primary prevention of the Developmental Origins of Health and Disease (DOHaD) [66,67,68,69]. The physiological state in which the parents conceive is universally accepted as a crucial factor in the prevention of pregnancy-related diseases and NCD-risk components [66,70]. Earlier interventions aiming at optimizing pre-gravid BMI and nutrient reserves may be of help inensuring a healthy pregnancy and inbreaking the intergenerational cycle of malnutrition and NCDs [68]. The optimum nutrient intake must be achieved prior to pregnancy to favor the overall reproductive and/or pregnancy cycle [66,68] as well as the health of the gut microbiota [61]. A recent systematic review indicated that preconceptual and pregnant women did not meet the recommendations for vegetable, cereal grain, folate, iron and calcium intake while the exceeding fat intake recommendations [71]. Dietary patterns that are higher in vegetables, fruits, whole grains, nuts, legumes, seeds and seafood andlower in red and processed meats and fried foods are associated with lower odds of preterm birth [72] and gestational hypertension [73]. Women with excessive pre-gravid BMI or overt diabetes must improve metabolic conditioning before pregnancy in order to decrease complications of fetal overgrowth [74], normalize nutrient levels and reduce the risk of congenital anomalies [53]. Maternal pre-gravid and early pregnancy metabolic condition programs early placental function and gene expression [51].

The ongoing initiatives, both at the individual level and the population level, aim to educate future parents in managing their own health for the future well-being of the next generation [62,75]. It is acknowledged that, to be successful, these programs need to comprise different sectors and disciplines [76,77] and focus on the promotion of health literacy during adolescence to be robustly addressed and implemented in multiple-integrated interventions, including programs on sexual health and contraception [70,78]. Multifaceted approaches used in strategies to reduce overweightness and obesity among children and young people in China, the Netherlands, the UK and the USA led to significant reduction in weight [77]. In particular, educational interventions should engage students with science knowledge and awareness on how their lifestyle impacts the health of their future children through interactive formats, i.e., experiential learning [75]. Education strategies must adapt to the preferred communication channels of adolescents by using motivational messages specifically tailored to this group in a dynamic and interactive mode. Media and digital technology offer new possibilities for engagement and service delivery [79,80,81,82]. Nutrition education must communicate clear, accurate and actionable messages with a specific goal for target groups. In this regard, more rigorous efforts for improving communication will become paramount to counteracting the information avalanche (i.e., too much information, countless competing messages coming from too many stakeholders), much of it incorrect, that we are all inundated with in today’s world and, finally, to enabling responsiveness to healthy guidelines or information [83]. The overall debate also recommends the responsibility of the private sector in NCDs’ prevention and control to prevent the harm caused by unhealthy commodity industries in terms of public regulation and market intervention to be appraised by national governments, along with non-governmental organizations, academics and civil society [84,85,86]. There is the explicit need to focus on interventions that address the food environment and impact obesity reduction [87]. Good governance and the rule of law have been seen as capable of advancing health and justice. Sugar, tobacco and alcohol taxes have helped in reducing harmful behaviors. Laws could ban trans-fat or excess saturated fat, salt and sugar; restrict the marketing of junk food to children; and require healthy school lunches. Subsidies could be directed to fruits and vegetables [88]. A meta-analysis showed that a 10% decrease in price increased the consumption of healthy foods by 12%, whereas a 10% increase in price decreased the consumption of unhealthy foods by 6% [77]. City planning can encourage physical activity, such asbike and walking paths, parks and playgrounds [88].

Nevertheless, obesity and pregnancy prevention and treatment strategies are often developed in a socially de-contextualized way and have seldom been successful in the long term. Ideally, government legislation should guarantee an “enabling environment” that provides opportunities, i.e., the chance for individuals to choose in a responsible manner. For the most part, action aimed at preventing adolescent pregnancy relies on measures focusing on changing the behavior rather than addressing the underlying determinants and drivers (i.e., poverty, social pressures, exclusion from educational and job opportunities and negative attitudes and stereotypes about adolescent girls). Working only in emergency situations cannot help. As highlighted by DOHaD researchers, patterns of adult disease are correlated with early life experiences that are socially embedded in existing structures of inequality such as social position, gender, race and ethnicity [89]. Namely, “bio-social capital” combines phenotypic health with social resources and care (i.e., neuro-cognitive and emotional skills, educational achievements, social networks and care including close family, relatives and friends, economic resources and contextual factors) [90]. To ensure a good start to life for every child, we must look at structural drivers of the problem, that is, to investigate the“causes of the causes” [91,92]. In accordance with the latest OECD Employment Outlook [93]: “*It is time to think big and address the right structural issue*”. There is a need for a whole-of-the-system way of thinking about the problem, acknowledging the implications of socioeconomic stagnation and growing inequality on society as a whole. There is research that shows that high inequality within a country leads to mental illness, lower life expectancy, higher infant mortality, obesity, children’s educational performance, adolescent births and growing problems of social trust and homicides [94]. A systems-based approach to obesity and pregnancy prevention may start with the community’s current systems and contexts by working collaboratively to understand the multilevel drivers of the problems and to identify ways that the existing systems can be used or reoriented to create better health [36,77]. It is also of crucial importance to engage adolescents themselves in developing policies and programs that relate to them and to work in equitable collaboration with them, as their perspective needs to be embedded into the democratic processes of decision making [3]. A growing number of young people are activists who highlight the effects of agriculture and the food industry on planetary ecosystems. They are playing increasing leadership roles in new social movements, which often combine local action with networking and campaigning through social media by targeting both individual responsibility and a system-level goal to change the system [42,77].

## 3. Is the Social Contract with Adolescents and Youth Broken?

According to Nobel prize-winning authors Abhijit Banerjee and Esther Duflo [95]: “*It is too hard to stay motivated when everything you want looks impossibly far away. Moving the goalposts closer may be just what the poor need to start running toward them*”, and “*A steady and predictable income makes it possible to commit to future expenditure and also makes it much easier and cheaper to borrow now…good jobs mean that children grow up in an environment where they are able to make the most of their talents*”. 

The future generations seem to effectively be the “lost generations”, devoid of hope for a bright future. Why has this come about? There are a number of interconnected drivers, namely: **Skills dilemma:**○Globotics: The life phase of work is increasingly characterized by a large number of stages combined with reorientation, entailing the ability to engage in life-long learning [96]. Skills are becoming redundant almost as soon as they are acquired. This is driven by globotics (globalization and robotics) squeezing the world of work simultaneously at the same breakneck pace [97]. There is also a mismatch with an archaic education system which no longer fitsthe career pathways possible or the skillsets needed for the job market. This skills obsolescence affects young people, their ability to work and their mental health, dealing with constant inevitable change. In the UK, the government is predicting that we will have six different careers in our lifetime, and 60% of those jobs do not yet exist [98].○NEETs: The number of NEETs (young people who are not in employment, education or training) hasbeen growing for some time now. In 2020, 22.4% of young people in the world were categorized as NEETs [99]. The COVID-19 pandemic contributed to an increase in the NEET rate to nearly 14% in Europe [100], and 28% in the USA [101] as a result of job losses and barriers to education and training. Young people who are neither in employment nor in education or training are at risk of becoming socially excluded. They are often left out of policymaking and are therefore particularly vulnerable to shocks, as we have seen with COVID-19. Furthermore, NEETs are radicalized in populations where youth unemployment or underemployment breeds social problems such as joining gangs (Americas) or radical groups (Middle East, North Africa and Afghanistan)**Work dilemma**: Young people face a lack of secure jobs and pension benefits that is incomparable with thatof previous generations. In most OECD countries, the levels of youth unemployment are twice as high as those of adults [102]. Young people are experiencing greater job losses and insecurity about their professional and financial futures. The impact of the pandemic on education is damaging young people’s opportunities to accumulate human capital. The suspension of schooling is likely to hinder skills formation while reinforcing inequalities between the most privileged and the most vulnerable.**Economic dilemma:** The costs of climate change obligations, COVID-19 and the Global Financial Crisis of 2008–2009 have all been pushed onto the next generation. Economic insecurity is rife: according to OECD reports, “*more than one in three people are economically vulnerable, meaning they lack the liquid financial assets needed to maintain a living standard at the poverty level for at least three months*” [103]. There is now a term for this: the “precariat” [104].**Social dilemma:**○Growing inequality: Nearly half (44%) of total gains from the globalization process have fallen into the hands of the richest 5% in the world, of which 19% has gone to the richest 1% [105]. In the United States, between 1976 and 2007, 58 cents of every dollar of real income growth went into the pockets of the top 1% of households [106]. ○Squeezed middle class: The middle class is being squeezed in many OECD countries, with standards of living stagnating or declining, and while the current generation is one of the most educated, it also has lower chances of achieving the same standard of living as its parents [103,107].Lack of social mobility: According to the OECD, “*The vicious confluence of poor educational opportunities, low skills and limited employment prospects can trap people in situations where they are also are far more likely to be exposed to environmental hazards and violence. As a result of this multidimensional inequality, while some individuals, cities and regions thrive, others fall further behind*” [108].

Dilemmas likely contribute to worsening of health by creating barriers to healthy environments, services and literacy. They are amplified among populations living in low-resource and low socioeconomic backgrounds. Young people in low socioeconomic status (SES) settings experience environments containing more cues for unhealthy behaviors (e.g., overeating, smoking, excessive drinking or physically inactivity), with psychosocial exposures potentially decreasing their executive functioning skills to resist those cues [109]. In particular, they are more and more exposed to obesogenic environments [8]. Gender inequality exacerbates all such conditions, making girls the most vulnerable. 

## 4. How the Dilemmas from the Broken Social Contract Drive the Problems of Adolescent Pregnancy and Obesity

Obese and pregnant adolescents are the result of diverse underlying societal, economic and other forces. Adolescent pregnancy can be associated with socioeconomic deprivation as both a cause and a consequence. Girls faces work, social and economic dilemmas, as well as gender inequality. The lack of job, services and education feeds into inequalities and amplifies vulnerabilities, unhealthy behaviors, insecurity and long-term implications for educational and occupational achievements within the parents–offspring pair. Around the globe, impoverished and poorly educated girls are more likely to become pregnant than their wealthier and educated counterparts; adolescents from an ethnic minority or marginalized group, who lack choices and opportunities in life or who have limited or no access to sexual and reproductive health are more likely to become pregnant [39]. Although the right to exercise choice is fundamental to the Universal Declaration of Human Rights, making “healthy choices” is difficult for members of the population who live in low-resource settings and who are socially, educationally, economically or politically disadvantaged [38]. Poverty and socioeconomic inequalities impact the rates of adolescent motherhood; this emerged from the analysis of results by World Bank income groups, suggesting an inverse relation between income and unintended pregnancy, resulting in substantial differences between low-income and high-income countries, with low-income countries having the highest unintended pregnancy rate and the lowest proportion of unintended pregnancies ending in abortion for 2015–19 [110]. Women and children in lower socioeconomic groups seem to show high levels of obesity compared with the rest of the population [24]. Findings from the Millennium Cohort Study, a nationally representative, ethnically diverse birth cohort looking at the trajectories of fat mass index and their ratio over 10 years across childhood and adolescence, showed a less healthy body composition among children and young people growing up in disadvantaged socioeconomic circumstances and living in more deprived areas compared with the more advantaged counterparts [111]. Mother–offspring dietary trajectories are stable across early life, with poorer diet quality associated with maternal socio-demographic and childhood adiposity [112]. Poverty likely predisposes low-income individuals towards a suboptimal diet, mostly due to limited purchasing power, which favors fast-food and convenience stores carrying shelf-stable junk foods whilst limiting the access to food outlets stocked with fresh products and whole grains. Nutrient-dense foods such as fresh fruits and vegetables may have a high “per calorie” cost compared to calorie-dense foods that are low in nutritional value [113]. Living in low SES families, with parents lacking social support or who are unemployed, belonging to a minority group or migrant background and experiencing adverse childhood conditions all lead to a greater risk of obesity [114]. Adolescent mothers are a population at academic and occupational risk. Moreover, NEET status is higher among females than among males [115,116]. The disconnection from the educational system and the workforce may increase the susceptibility for risky health behaviors, which arealready at higher rates among adolescents, ultimately impacting future health [101]. The educational and occupational attainments of parents are basic factors for the family’s SES, with a higher SES linked to more resources for children’s academic and career development and a larger social context for youth competence development, which may predict young adults’ educational and occupational achievements [117]. Parents’ education is a form of capital with positive implications for youth achievement. Health literacy may help create a pro-active environment building health, educational and job opportunities. The relevant abstention from formal education among pregnant adolescents causes many of them to not have the opportunity to continue their education once they become a mother. It prevents girls from using their potential to invest in education, find a job and have an income and increases the chances of a recurring pregnancy [118]. Growing job insecurity and the systematic labor market exclusion of youth at the very beginning of their professional careers are likely linked with high risks of poverty, being part of the precariat, social exclusion, disaffection and insecurity and a higher propensity toward unhealthy behaviors [119,120]. The loss of education and unemployment—that is, the lack of food skills and financial resources, respectively—may lead to increased food insecurity [100], with consequences on the health of the mother–child dyad.

Emerging from the arguments herein discussed, obese adolescent pregnancy can be considered a marker of the broken social contract (Figure 2). 

## 5. Conclusions and Future Directions

Adolescent obesity and pregnancy are of great concern because of the long-lasting consequences on the health and well-being of both the parents-to-be and the next generation. Beyond medical risks, there is a raised social risk related to the physical, emotional, economic and social dependence of adolescent mothers, which may jeopardize several aspects of the mother/infant’s future life. Limited perspectives of personal growth, a lack of educational incentives, the labor market and healthy environments affect their opportunities and expectation for the future. Persistent obesogenic environments oppose efforts to facilitate weight loss and prevent weight regain. An obese “precariat” is likely to be in a perpetuating cycle, creating health, economic and social problems that will not just go away.

Considering that the social contract with young people has several break points, society is impelled to reattain the trust of adolescents. Further efforts on the part of policymakers, healthcare providers and the community must be oriented towards guaranteeing equity and healthy environments for today’s adolescents by implementing the key-factors of successful interventions whilst removing barriers. A more holistic approach is paramount to achieving sustainable, permanent solutions in the real world. This implies a broader perspective ofpolicymakers, including changes in the environment where adolescents live. Prevention cannot be a matter of individual “lifestyle” choices only but requires collective and policy actions to move from short-term direct benefits or risks to longer-term collective benefits or costs, including the engagement with the food industry. Early-life nutrition, diet diversity, food environments and socioeconomic factors must be considered as the basis on which tofurther scaleup the ongoing initiatives. Actions must be delivered through platforms both within and outside health facilities (e.g., community-health workers, schools and mass media) and cross-cutting sectors to nutrition—notably, education, food systems and social safety nets. The health profession may have a key voice in advocating for law taxation and regulation related to food environment and climate. Personal knowledge and competencies need to be mediated by organizational structures and the availability of resources, which enable people to access, understand, appraise and use information and services in ways that promote and maintain good health and well-being for themselves and those around them. Adolescent interventions must vary in terms of form and effectiveness by age, sex, income level and geography, along with social, cultural and country context. Some interventions must be adolescent-specific, and others must be adolescent-inclusive. It is worth remembering that achieving universal health coverage requires adolescents themselves to be empowered to initiate action and influence decisions that affect their health and development through meaningful participation. Although personal behaviors are influenced by environments, people can act as agents of change in their roles as citizens. Young people themselves are demanding roles beyond being consumers. Adolescents have the potential to unlock the political and policy paralysis around unhealthy and unequal environments and systems. Stronger participation and demand from civil society for political action arenecessary.

## Figures and Tables

**Figure 1 nutrients-14-03550-f001:**
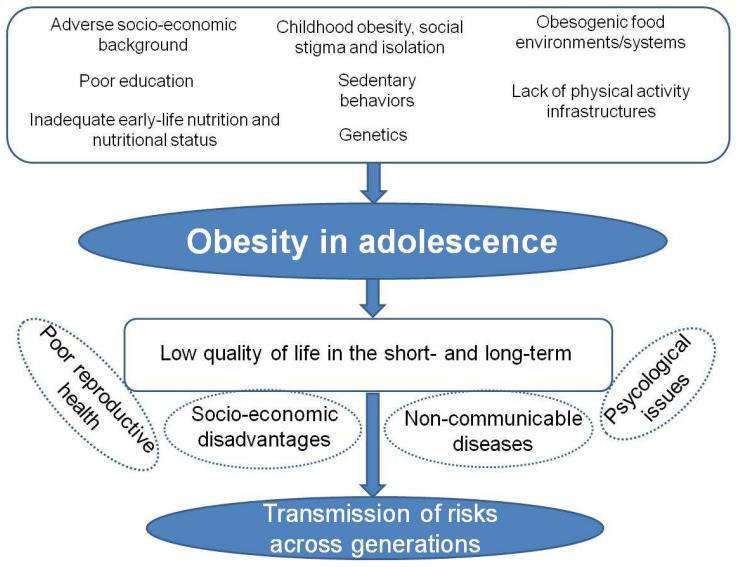
The interconnected main risk factors and the short- and long-term consequences of obesity in adolescents.

**Figure 2 nutrients-14-03550-f002:**
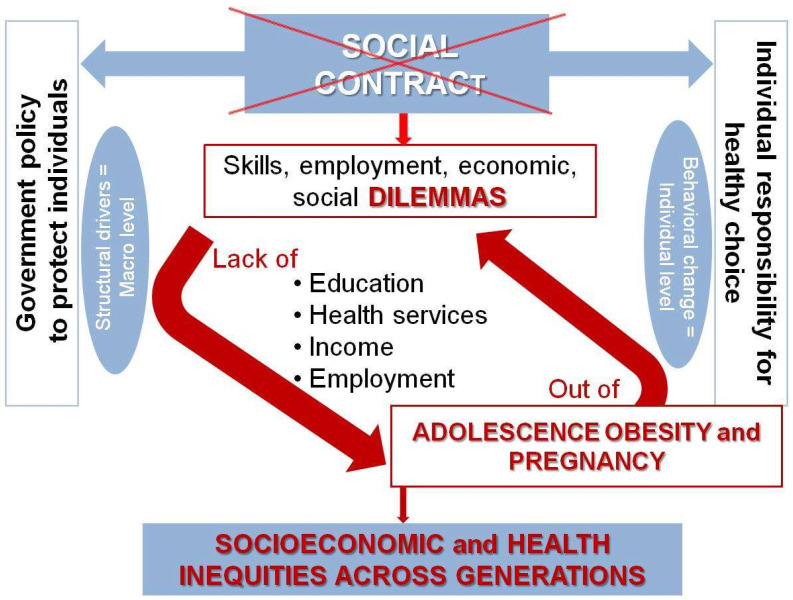
The link between the broken social contract with youth and pregnant adolescents with obesity: a vicious cycle of deepening structural drivers exacerbating the situation across generations unless the policy interventions actively create transformative change.

**Table 1 nutrients-14-03550-t001:** The short- and long-term adverse implications forthe mother–child dyad’s health associated with maternal excessive weight.

	Adverse Implication
** *Mother* **	***Early in pregnancy:*** Increased risk of spontaneous abortion and congenital anomalies [48].
	***In later pregnancy:*** Increased risk of gestational diabetes, hypertensive disorders, cardiac dysfunction, proteinuria, sleep apnea and non-alcoholic fatty liver disease [49,50].
	***At delivery:*** Increased risk of instrumental and cesarean birth, surgical site infection and venous thrombosis [48].
** *Offspring* **	Increased risk for fetal overgrowth, mortality and morbidity [49,51].
	Increased risk for neonatal morbidity and later obesity and metabolic syndrome [49,52]. Increased risk of premature death among adult offspring [53].

## Data Availability

Not applicable.

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
