# Peer review of "Obesity, Pregnancy and the Social Contract with Today’s Adolescents"

_nutrients, 2022, doi:10.3390/nu14173550_

Round 1
Reviewer 1 Report
Thank you for inviting me to review the following article. I have some questions:
1- The manuscript seems outside the scope of the special edition for which it was submitted. The focus is very low on the type of maternal nutrition and outcomes for the offspring.
2- It is not clear in the introduction what the purpose of the article is. There is also no clear research question.
3- The methodology addressed in the article is not clear.
4- The article seems to start from a conceptual model involving the term social contract, to deny it, but this is also not clear from the beginning of the article.
5- Teenage pregnancy and the obesity associated with such pregnancy is treated in the article as a problem exclusively for young pregnant women.
Isn't weight gain, and maintaining a high BMI, a phenomenon that occurs in women of all ages?
6- Dilemmas are treated as generalized for all countries in the world.
However, not all countries have problems related to the demographic dilemma, for example. Dilemmas are more relevant depending on the country or geographic region of the globe analyzed.
7- Most of the problems presented are problems for young people and adults. It is not clear, for all the problems presented, why they are greater for young people.
8- The article is pessimistic and suggests a bad future with problems passed down through generations.
If 60% of the jobs that people will occupy do not yet exist, how to define a future with the same problems as the present.
9- The solution points out by the authors in the conclusion to the problem of pregnancy and obesity in youth is broad.
It does not provide examples of concrete changes that have been made that have reduced the number of pregnancies and obesity among young people.
It is also unclear how such changes are more effective than changes in individual and lifestyle choices. Without such individual changes, any other obesity-related solutions are palliatives.
10- Some dilemmas presented have always existed. For example, economic and social dilemmas have always existed. Is there no solution for the future? The article is a bit speculative.
Reviewer 2 Report
Dear Authors,
Thank you for allowing me to review your manuscript "The horns of the dilemma: obesity, pregnancy and the broken social contract with today’s adolescents". I find this study to be a well-argued review, an overview of challenges to adolescent global health. I enjoyed reading it.
The strength of this article is the concept of a social contract between adolescents and the rest of society as a framework for thinking. The authors paint with broad "advocacy" strokes, making the text pleasant to read. In some parts, however, I find the text too rhapsodic. To me, the references and background information on obesity could benefit from more detail and deeper analysis. For example, although a person can undoubtedly get pregnant overnight, obesity does not develop overnight. The analysis and figure 1 lack factors that contribute to the continuous development of obesity for genetically susceptible individuals from childhood, where the largest risk factor for adolescent obesity would be childhood obesity (Simmonds et al), the associated stigma, and the lack of appropriate prevention and health care to curb the development of obesity.
Many of these factors would be exactly those that mirror social vulnerability and also increase the risk of early pregnancy.
Similarly, the interesting framework of a social contract can be developed. A contract has two parts. The adolescents' part - what is it? The large cohorts of individuals who "did their part", for example, got an education, only to find that a person needs two Masters and a BA to get a position also in a low-income country, did they do their part? What do we expect from the adults-to-be? It would be interesting to hear the authors discuss that. The climate change and legacy we leave behind, it that a reason for despair leading to obeisty and early pregnancy? Or are we just looking at a move from a global focus on childhood survival to concern about the fate of large groups of adolescents?
The discussion on the balance between the individual´s responsibility for making healthy choices and the legislative processes of a country is also a bit tendentious and can be developed. Sugar-taxes, tobacco laws, and similar controversies illustrate the problems.
Having said that, I still think the article should focus on the obesity+pregnancy combination.
Minor comments:
Title: The idiom "horns of a dilemma" represents a choice between two things, both unpleasant. Clearly, this article does not argue for choosing pregnancy OR obesity. Instead, your work illustrates the global trends that affect both phenomena. I find the title, though witty, not underpinned by the article.
Abstract: Nice. "Dual challenge" seems the right approach.
Page 3: I find the "In fact", and "actually" sounding a bit - well, surprised. These are just facts. I also dislike the slash and prefer "overweight or obesity". There are good references about psychosocial wellbeing among adolescents with severe obesity.
Page 6: Add: For example, the OECD average DEPENDENCY RATIO was 13.9 in 1950, 22.5 in 2000, rising to 35.2 in 2025 and 53.2 in 2050
Page 6: Taxes are resources. Change into "resources and care"
Page 8: "Unhealthy and pregnant adolescence" sounds a bit strange and seems larger than obeisty and pregnancy. The article should probably stick to the obesity and pregnancy dual challenge.
Page 8. This sentence sounds strange: "Mostly, adolescent pregnancies are usually the result of an absence of choices". Most pregnancies are likely the result of sexual intercourse. If you mean the right to abortion, please say to.
Conclusion: please elaborate a bit on the contribution by adolescents themselves.
Yours respectfully,
R
Round 2
Reviewer 1 Report
Thank you for the opportunity to review the paper.
The authors made efforts and improved the article.
I consider the article accepted in its current format.